# Effects of Five Prebiotics on Growth, Antioxidant Capacity, Non-Specific Immunity, Stress Resistance, and Disease Resistance of Juvenile Hybrid Grouper (*Epinephelus fuscoguttatus* ♀ × *Epinephelus lanceolatus* ♂)

**DOI:** 10.3390/ani13040754

**Published:** 2023-02-19

**Authors:** Li Zhu, Shaoqun Wang, Yan Cai, Huizhong Shi, Yongcan Zhou, Dongdong Zhang, Weiliang Guo, Shifeng Wang

**Affiliations:** 1State Key Laboratory of Marine Resource Utilization in South China Sea, Hainan University, Haikou 570028, China; 2Hainan Provincial Key Laboratory for Tropical Hydrobiology and Biotechnology, College of Marine Science, Hainan University, Haikou 570228, China

**Keywords:** hybrid grouper, prebiotics, growth performance, immunity, stress resistance, disease resistance

## Abstract

**Simple Summary:**

Prebiotics, as feed additives, have attracted much attention in aquaculture. Generally speaking, different prebiotics have different effects on fish. This manuscript describes the effects of five prebiotics on growth, antioxidant activities, non-specific immunity, stress resistance, and disease resistance of juvenile hybrid grouper (*Epinephelus lanceolatus* ♀ × *Epinephelus fuscoguttatus* ♂). We found that among the five prebiotics, mannan oligosaccharides and xylooligosaccharides showed the best overall effects in improving the growth performance, non-specific immunity, stress resistance, and disease resistance of pearl gentian grouper. This conclusion provides the theoretical basis for the prebiotics application to guide the healthy development of aquaculture.

**Abstract:**

To explore the short-term health benefits of five prebiotics on hybrid grouper (*Epinephelus fuscoguttatus* ♀ × *Epinephelus lanceolatus* ♂), six experimental groups fed with different diets (basal diet, diet control (CON); basal diet + 0.2% fructooligosaccharide (FOS), diet FOS; basal diet + 0.5% chitosan, diet chitosan (CTS); basal diet + 0.2% mannan–oligosaccharide (MOS), diet MOS; basal diet + 0.1% β-glucan (GLU), Diet GLU; basal diet + 0.05% xylooligosaccharide (XOS), diet XOS) were set up, and a 4-week feeding trial was conducted. MOS and XOS significantly improved the growth of hybrid grouper compared to the CON group (*p* < 0.05). Antioxidant enzyme assay showed that the activity of glutathione peroxidase (GPx) was significantly enhanced in the MOS group, and the content of malondialdehyde (MDA) in the XOS group was significantly lower than in the CON group (*p* < 0.05). The catalase (CAT) activities were significantly enhanced in all prebiotic-supplemented groups compared with the CON group (*p* < 0.05). Non-specific immunity assay showed that the activities of alkaline phosphatase (AKP) and lysozyme (LZM) were significantly increased in all prebiotic-supplemented groups compared with the CON group (*p* < 0.05). The total protein content in the XOS group was significantly increased (*p* < 0.05), and the albumin (ALB) activity in the MOS group was more significantly increased than that in the CON group. Histological examination of the intestine revealed that muscle thickness was significantly increased in all prebiotic-supplemented groups compared to the CON group (*p* < 0.05). Villi length, villi width, muscle thickness all increased significantly in the MOS group (*p* < 0.05). In addition, the crowding stress and ammonia nitrogen stress experiments revealed that the survival rates of the MOS and XOS groups after stresses were significantly higher than those of the CON group (*p* < 0.05). Though MOS and XOS exhibited similar anti-stress effects, the antioxidant and non-specific immunity parameters they regulated were not the same, indicating that the specific mechanisms of MOS and XOS’s anti-stress effects were probably different. After being challenged with *Vibrio harvey*, MOS and GLU groups showed significantly higher post-challenge survival rates than the CON group (*p* < 0.05). These findings indicated that among the five prebiotics, MOS and XOS showed the best overall short-term beneficial effects and could be considered promising short-term feed additives to improve the stress resistance of juvenile hybrid grouper.

## 1. Introduction

The hybrid grouper, commonly named pearl gentian grouper, was produced by fertilizing the eggs of brown-marbled grouper (*Epinephelus fuscoguttatus*) with the sperm of giant grouper (*Epinephelus lanceolatus*). Characterized by rapid growth, high-quality meat, and good market price, hybrid grouper is a widely cultured grouper breed in the coastal areas of Southeast Asia [1,2,3,4]. As with many other aquaculture animals, groupers may encounter various stressful situations, such as water quality deterioration after stormy weathers, crowding/oxidative stresses during catching/transportation, or seasonal epidemic after extreme temperature fluctuation [5,6,7]. The hybrid grouper needs strategies for increasing fish stress resistance against these unfavorable conditions and pathogens. Functional dietary supplements are one of the solutions to effectively enhance cultured fish’s resistance to varied stresses [8].

Prebiotics are non-digestible food ingredients that selectively promote or inhibit the growth/activity of certain microorganisms in the gastrointestinal tract, therefore producing benefiting effects to the host [9]. Comparing to antibiotics, prebiotics do not induce drug resistance or environmental pollution. Studies demonstrated that using prebiotics as feed additives has a great potential to become a valid alternative method for improving fish health [10,11]. The most commonly used prebiotics in aquaculture includes fructooligosaccharides (FOS), β-glucan (GLU), chitosan (CTS), mannan–oligosaccharides (MOS), and xylooligosaccharide (XOS) [12,13,14,15]. FOS supplementation in fish feed can promote growth performance, feed efficiency, immune response, and disease resistance in fish species, such as stellate sturgeon (*Acipenser stellatus*), turbot (*Scophthalmus maximus*), and blunt snout bream (*Megalobrama amblycephala*) [16,17,18,19,20]. As an immune stimulant, GLU can trigger the immune system of aquatic animals by binding to macrophages, neutrophils, and specific receptors on macrophages [21,22,23]. CTS supplementation significantly improves the phagocytosis activity of macrophages and the antioxidant defense system activity of tilapia [24]. Dietary MOS can significantly enhance the gut morphology of gilthead sea bream (*Sparus aurata*), rainbow trout (*Oncorhynchus mykiss*), and hybrid grouper (*E. fuscoguttatus* ♀ × *E. lanceolatus* ♂) [25,26,27]; increase the growth performance of grass carp (*Ctenopharyngodon idella*); improve the feed conversion ratio of common carp (*Cyprinus carpio*) [28,29]; and improve the disease resistance and antioxidant capacity of hybrid grouper (*E. fuscoguttatus* ♀ × *E. lanceolatus* ♂) and grass carp (*C. idella*) [27,29,30]. XOS, as a new prebiotic, has shown beneficial effects on both skin and intestine mucosal immunity of Caspian white fish (*Rutilus kutum*) [31]. Generally speaking, different prebiotics have different effects on fish. Many factors, such as prebiotic structure, dosage, supplementation period, fish species, and fish age/stage/weight, might contribute to the final results of prebiotic supplementation [28,32]. It is also noteworthy that prebiotics, like other means of supplementation, have their limitations. It should be used carefully during specific periods of fish life because medium/long-term use of prebiotics might result in adverse effects in fish growth/health as well as unnecessarily increasing the production costs. Therefore, research regarding comparison of various prebiotic’s specific beneficial effects is necessary to identify the optimal prebiotics which could be applied prior to specific predictable stressful conditions (such as heavy rainfall, fish catching/transportation, or heat waves).

Previous studies of hybrid grouper mainly focused on the nutritional aspects of this fish, such as protein and amino acid, lipid and essential fatty acid, alternative protein source, mineral nutrition, etc. [33,34,35,36,37]. Studies regarding prebiotics’ effects on this species were few. Besides two MOS studies in hybrid grouper (both from our lab) [27,38], there is limited information concerning how other prebiotics might affect this fish species. Therefore, in this study, the effects of five prebiotics as feed additives on the growth performance, antioxidant capacity, non-specific immunity, resistance to ammonia nitrogen/crowding stresses, and resistance to pathogen infection in hybrid grouper were evaluated. The results of this study provide references for the potential use of prebiotics FOS, GLU, CTS, MOS, and XOS in hybrid grouper aquaculture.

## 2. Materials and Methods

### 2.1. Diet Preparation

A commercial grouper feed (Guangdong Hengxing Aquatic Products Co., Ltd., Zhanjiang, China) was used as the basal diet. The proximal composition of the basal diet includes: crude protein, 49.0%; crude lipid, 9.0%; crude fiber, 2.0%; crude ash, 16.0%; total phosphorus, 1.8%; moisture, 10.0%; and lysine, 2.5%. Six experimental diets (corresponding to 6 experimental groups) were prepared: basal diet (group CON), 0.5% chitosan (group CTS, 5 g chitosan + 995 g basal diet), 0.1% β-glucan (group GLU, 1 g β-glucan + 999 g basal diet), 0.2% fructooligosaccharides (group FOS, 2 g fructooligosaccharides + 998 g basal diet), 0.2% mannooligosaccharides (group MOS, 2 g mannooligosaccharides + 998 g basal diet), 0.05% xylooligosaccharides (group XOS, 0.5 g xylooligosaccharides + 999.5 g basal diet) (Table 1). All prebiotics were purchased from Shandong Shengyuan Biotechnology Co., Ltd., Qingdao, China). Supplementation levels of 0.5% CTS, 0.1% GLU, 0.2% FOS, 0.2% MOS, and 0.05% XOS were chosen based on optimized results from previous works of the same or closely related species. For all five prebiotics, wide dose ranges of its supplementation had been studied [29,38,39,40,41,42,43,44,45]. To prepare prebiotic-supplemented diets, each prebiotic was first dissolved in 25 mL of sterile purified water, then the solution was sprayed evenly onto the commercial pellets. For the control diet, 25 mL of sterile purified water was sprayed onto the commercial pellets. New batches of experimental diets were prepared every week to ensure the quality of the experimental diets. All sprayed feed was air-dried at 25 °C and stored in plastic bags at 4 °C for further use.

### 2.2. Experimental Fish and Sample Collection

Experimental grouper (27.48 ± 0.63 g initial body weight, IBW) was purchased from Yu Hai Lan Ke Biotechnology Co, Danzhou, China. During the two weeks of acclimation, the fish were fed a basal diet. After acclimation, they were randomly distributed into 18 cylindrical tanks (each containing 300 L water) at a stocking density of 35 fish per tank. Each experimental group included 3 tanks. Each group was fed one of six experimental diets twice daily (09:00 and 17:00) at 4% body weight (BW) per day. The duration of the feeding experiment was 4 weeks. Throughout acclimation and the feeding trial, fish were maintained in a flow through seawater system (salinity at 33‰; flow rate at 0.2 L/min). The running aerated seawater quality was maintained as: pH at 8.0 ± 0.2, dissolved oxygen at 5.8 ± 0.3 mg L^−1^, ammonia nitrogen at lower than 0.2 mg L^−1^, and nitrite at lower than 0.05 mg L^−1^.

After the feeding experiment ended, feeding of fish stopped for 24 h before sampling. Nine fish per group (3 fish per tank) were randomly selected and anesthetized with MS-222 at a concentration of 0.1 g/L. After weighing, the body surface of sampled fish was first sterilized with alcohol cotton balls. Then, blood samples were collected from the caudal vein using 1 mL sterile syringes. The sampled blood was transferred into sterile centrifuge tubes and allowed to stand for 3 h at room temperature and placed overnight at 4 °C. Blood samples were centrifuged (4 °C, 1500 rpm, 15 min), and serum was separated and stored at −80 °C to analyze serum antioxidant and non-specific immunity parameters. After blood withdrawal, each sampled fish was dissected using sterilized scissors and forceps. Two cm section of the intestines (midgut) were removed and preserved in 4% paraformaldehyde for histological analysis. All animals’ treatments were in accordance with the guidelines of the Animal Experiment Ethics Committee of Hainan University, Haikou, China.

### 2.3. Growth Performance

All fish were weighed at the beginning and end of the feeding trial (Day 0 and Day 28). Weight gain rate (WGR), specific growth rate (SGR), and food coefficient (FC) were calculated according to the following formulae: WGR (%) = [(Wfin − Win)/Win] × 100, SGR (%) = [ln Wfin − ln Win]/ d × 100, FC = FI/(Wfin − Win). Where Wfin is the final mean weight, Win is the initial mean weight, and d is the duration of feeding (days).

### 2.4. Serum Parameter Analysis

The serum samples stored at −80 °C were placed on ice to thaw spontaneously.

#### 2.4.1. Antioxidant Activity

Superoxide dismutase (SOD), catalase (CAT), glutathione peroxidase (GSH-PX), and malondialdehyde (MDA) content in the serum were determined using commercial kits (Nanjing Jiancheng Bioengineering Institute, Nanjing, Jiangsu, China) according to manufacturer’s instructions. The SOD activity (Cat. No. A001-3-1) was measured by the WST-1 method; the CAT activity (Cat. No. A007-1-1) was measured by the visible light method; the GSH-PX activity (Cat. No. A005-1-2) was measured by the colorimetric method; and the MDA content (Cat. No. A003-1-2) was measured by the TBA method. All reactions were performed in 96 wells, and the absorbance was read by a microplate reader (Thermo Scientific™ 5580, Shanghai, China).

#### 2.4.2. Non-Specific Immunity Activities

Lysozyme (LZM), alkaline phosphatase (AKP), albumin (ALB), and total protein (TP) content for serum were measured using commercial kits (Nanjing Jiancheng Bioengineering Institute, Nanjing, Jiangsu, China) according to the manufacturer’s instructions. The LZM activity (Cat. No. A050-1-1) was measured by the turbidimetry method, the AKP activity (Cat. No. A059-2-2) was measured by the microenzyme labeling method, the ALB activity (Cat. No. A028-2-1) was measured by the Bromocresol green method, and the TP content (Cat. No. A045-2-2) was measured by the Coomassie Brilliant Blue method. All reactions were performed in 96 wells, and the absorbance was read by a microplate reader (Thermo Scientific™ 5580, Shanghai, China).

### 2.5. Histological Examination of the Intestine

At the end of the 28-d feeding trial, intestinal samples (midgut) from three fish per tank (9 fish per group) were collected and preserved in 4% paraformaldehyde [46]. The fixed samples were sent to Seville Biotechnology Co., Ltd.(Wuhan, China) for paraffin sectioning, staining, and photographing. Intestinal villus height, intestinal villus width, and thickness of tunica muscularis were measured and recorded using a microscope (Olympus, DP72, Tokyo, Japan) and CaseViewer 2.3 software (3DHISTECH Ltd., Budapest, Hungary). For the measurements, 20 images per individual (4 quadrants/section × 5 sections/individual) were captured and measured. Each intestinal histological parameter was calculated based on 20 measurements per individual.

### 2.6. Ammonia Nitrogen Stress Experiment

At the end of the 28-d feeding trial, 20 fish in each group were randomly selected and redistributed evenly into three glass tanks (10 L each) for ammonia nitrogen stress tests. Fish were starved for 24 h before the ammonia nitrogen stress experiment began. The stock solution for ammonia in each tank was 1.0 mg/L of NH_4_Cl obtained by a series of preliminary experiments. The ammonia nitrogen concentration was continuously detected by the reagent method to keep the level of total ammonia nitrogen at 1.0 mg/L (pH = 6.8 ± 0.2; water temperature = 24 ± 1 °C). During the stress test, water temperature, pH level, and ammonia nitrogen concentration were determined every 2 h, and the concentration of non-ionic ammonia was adjusted using ammonium chloride (NH_4_Cl) solution. Mortality was recorded every 12 h during the period of 96 h stress test, and dead fish were removed. The cumulative survival rate (CSR) was calculated as follows:CSR (%) = 100 × number of survival fish after the stress test/number of fish before the stress test.

### 2.7. Crowding Stress Experiment

At the end of the 28-d feeding trial, 20 fish from each group were randomly selected and divided evenly into three glass tanks (10 L each) for the crowding stress experiment. Only 3 L of water was added to each tank to keep the fish density at 100 g/L. The crowding stress experiment lasted 24 h. During the experiment, the water temperature, pH, and dissolved oxygen (DO) were consistent with those during the feeding period. The mortality of each group was recorded at the end of the experiment, and the cumulative survival rate (CSR) was calculated as follows:CSR (%) = 100 × number of survival fish after the stress test/number of fish before the stress test.

### 2.8. Challenge Test

*V. harveyi* was previously isolated by our laboratory from orange-spotted grouper (*Epinephelus coioides*) [47,48]. It was preserved in tryptic soy broth (TSB) with 25% (*v/v*) glycerol and stored at −80 °C until use. The isolates were inoculated in Marine Broth 2216E medium with constant shaking at 30 °C for 24 h. Following 24 h of growth, the cultures were centrifuged at 4000× *g* for 10 min at 4 °C. Cultures were adjusted to optical densities of 0.02 at 600 nm by dilution with phosphate-buffered saline (PBS). Following adjustment, a sample of culture was taken, and viability was tested by bacterial plate count method (in duplicate) to determine the colony forming unit (CFU) present for the challenge. At the end of the 28-d feeding trial, ten fish from each tank were randomly selected for the challenge test. All fish were injected intraperitoneally with 0.1 mL bacterial suspension at lethal dose 50 (LD_50_) (1 × 10^7^ CFU/mL) [47]. Fish were carefully monitored, and mortality was recorded twice daily for 72 h. The survival rate was calculated. *V. harveyi* was re-isolated from dead fish and confirmed with standard biochemical methods (as described in Bergey’s Manual of Systematic Bacteriology) and the 16S rDNA sequencing method.

### 2.9. Statistical Analysis

All data were calculated as means ± standard errors of the mean (SEMs). A one-way analysis of variance (ANOVA) and a multiple comparison (Duncan) test were conducted to compare the significant differences using the SPSS 20.0 program (SPSS Inc., Chicago, IL, USA). *p* < 0.05 was considered statistically significant.

## 3. Results

### 3.1. Growth Performance

Figure 1 shows the growth performance of fish fed with different prebiotics. The WGRs and SGRs of the five prebiotic groups were higher than those of the CON group, but only MOS and XOS groups were significantly higher (*p* < 0.05) (Figure 1A,B). The food coefficients of the five prebiotic feeding groups were all lower than those of the CON group, but only the MOS group was significantly lower (*p* < 0.05) (Figure 1C).

### 3.2. Antioxidant Activities

Among the five prebiotics, only MOS significantly increased the GSH-PX activity (*p* < 0.05) of hybrid grouper (Figure 2A). The MDA contents of the XOS and FOS groups were significantly lower than that of the CON group (*p* < 0.05) (Figure 2B). The CAT activities of all prebiotic groups were significantly higher than that in the CON group (*p* < 0.05) (Figure 2C). The SOD activities of all prebiotic groups were not significantly different from that of the CON group (*p* > 0.05) (Figure 2D).

### 3.3. Non-Specific Immunity Activities

All five prebiotics significantly increased the AKP and LZM activities (*p* < 0.05) of hybrid grouper (Figure 3A,B). None of the five prebiotic groups showed significantly different TP content than the CON group (*p* > 0.05). The ALB activities of all prebiotic groups were enhanced to a certain extent compared to those of the CON group. Only the MOS group was significantly different from the CON group (*p* < 0.05).

### 3.4. Histological Examination of the Intestine

Intestinal morphology examination results showed that the length of intestinal villi in groups CTS, FOS, MOS, and XOS increased significantly (*p* < 0.05) compared to those of the CON group, and the width of intestinal villi in group MOS increased significantly compared to that of CON group (*p* < 0.05). Additionally, the thickness of muscle in group CTS, GLU, MOS, and XOS increased significantly in comparison with the CON group (*p* < 0.05) (Table 2 and Figure 4).

### 3.5. The Cumulative Survival Rate of Different Groups after Ammonia-Nitrogen Stress

The results of the 96-h ammonia–nitrogen stress experiment showed that compared with the CON group, MOS and XOS groups showed significantly increased survival rates of hybrid grouper (*p* < 0.05) (Figure 5). In contrast, the FOS, GLU, and CTS groups showed significantly decreased survival rates (*p* < 0.05). The survival rate of the XOS group was 100% after 96-h ammonia–nitrogen stress.

### 3.6. The Cumulative Survival Rate of Different Groups after Crowding Stress

The results of the crowding stress experiment showed that the survival rates of the MOS and XOS groups after 24 h crowding stress were significantly higher than those of the CON group (*p* < 0.05) (Figure 6). The survival rates of the FOS, GLU, and CTS groups were not significantly different from that of the CON group.

### 3.7. The Cumulative Survival Rate of Hybrid Grouper after V. harvey Infection

Compared with the CON group, the cumulative survival rates of the MOS and GLU groups were significantly higher than that of the CON group (*p* < 0.05) (Figure 7). The MOS group had the highest survival rate and the best protection effect against *V. harvey* infection. The cumulative survival rates of the FOS, CTS, and XOS groups were significantly lower than those of the CON group.

## 4. Discussion

In the present study, MOS and XOS supplementation significantly enhanced the growth of hybrid grouper, which is in accordance with Lee et al. (2018)’s study of MOS’s growth-promoting effect in Japanese eel (*Anguilla japonica*) and Li et al. (2008)’s study of XOS’s growth-promoting effect in juvenile turbot (*Scophthalmus maximus*) [49,50]. Although prebiotics’ growth-promoting effects are closely related to supplementation duration and dosage, longer duration or higher dosage does not always correspond to better growth performances. In GLU, longer supplementation duration seemed to correlate to better growth-promoting effects. For example, within five GLU supplementation studies (of the same dosage), studies with supplementation period of 2–4 weeks did not obtain significant growth promoting results in fish ([44,51], the present study), yet studies with GLU supplementation period of 6 weeks obtained significantly increased growth [41,52]. On the other hand, in MOS, longer supplementation duration might not correlate with better growth effects. Though 4 weeks of MOS supplementation resulted in significant improved growth in juvenile hybrid grouper [the present study], 9 weeks of MOS supplementation (at the same dosage) did not exhibit any growth-promoting effects [27]. Similarly, Grisdale-Helland et al. (2008), Dimitroglou et al. (2010), and Zhou et al. (2010) also observed non-significant differences in fish growth after MOS supplementation of 8 to 16 weeks. However, disparities did exist [26,53,54]. In a review of fish MOS studies, within the 11 studies with significant growth promoting results, 10 studies used a supplementation period ≥ 8 weeks, while only 1 study used a supplementation period less than 8 weeks [55]. 

Dosage is an important factor that influences the prebiotic effects. Based on a review of MOS supplementation effects, among the nine studies with significant growth-promoting effects, seven studies used an MOS supplementation dose ranging between 0.15–0.4%, whereas only two studies used an MOS dose above 1% [55]. Fish age/stage/weight is another factor that might be related to the effects of prebiotics. For instance, in European sea bass (*Dicentrarchus labrax*), when supplemented with the same dose (0.4%) and period (60 days) of MOS, fish with initial weight of 44.95 ± 2.99 g showed a significant increase in growth (*p* < 0.05) [56], whereas fish with initial weight of 116 g did not (*p* > 0.05) [57]. Many factors, such as prebiotic structure, dosage, supplementation period, fish species, and age, might contribute to the final results of prebiotic supplementation. The limited number of studies concerning the effect of various prebiotics in fish constrained our understanding of the function modes of prebiotics. In the future, more serial, in-depth research of prebiotics in representative fish species is needed to clarify the best supplementation strategy of each specific prebiotic in each fish species. 

Oxidative stress results from the overproduction of reactive oxygen species (ROS) [58]. Many unfavorable rearing conditions (e.g., high ammonia nitrogen, crowding, and low oxygen) could induce oxidative stress, which results in multiple damages, such as functional loss, apoptosis, or necrosis of cells [59]. MDA is one of the main products of lipid peroxidation induced by ROS [60]. SOD, together with CAT and GSH-PX, formed the major enzyme defense mechanism against the damaging effects of ROS [61,62]. The measurement of these parameters can provide an indication of the antioxidant status of fish. In the present study, all prebiotic-supplemented groups had increased CAT activities, but MDA was only reduced in FOS and XOS groups. The result indicated that CAT was not always negatively correlated with MDA. This is consistent with many other studies [63,64,65,66]. The explanation is rather obvious; after all, CAT is just one of numerous antioxidant enzymes that could prevent the production of MDA. Besides antioxidant enzymes, a large number of other antioxidants, such as non-enzymatic molecules (e.g., vitamin C, E) and certain prebiotics/probiotics, could also affect the final MDA level of fish [67,68,69]. In the present study, FOS, GLU, CTS, MOS, and XOS all displayed various levels of antioxidant effects, which was in accordance with many previous prebiotic studies [27,38,40,45]. The mechanism of the actions of these prebiotics generally involved at least one of the following aspects: the prebiotic promoted the growth of antioxidant bacteria [70,71]; the prebiotic was a strong in vitro antioxidant itself [72,73,74,75]; the prebiotic could bind to receptors in the intestine and exert antioxidant effects through the receptor-related pathways [56,75,76,77,78]. For example, the effects of FOS/GLU on SOD, CAT, and GSH-PX activities were associated with improved *Bacillus licheniformis/Bacillus subtilis* growth [70,71]. It was also reported that ß-glucan particles induced an oxidative burst by recognizing specific receptors [76,77]. Chitosan is a good antioxidant itself since chitosan can chelate metal ions or scavenge free radicals through the donation of hydrogen or one pair of electrons [72,73]. As for MOS, on the one hand, MOS could react with ROS (like hydroxyl radical) [74]. On the other hand, MOS can also activate the mannose receptor (MR) in fish, thereby up-regulating the antioxidant enzyme gene transcription [56,78]. Similar to MOS, XOS could exhibit strong in vitro antioxidant abilities and form conventional hydrogen bonds [75]. Although many studies have indicated the positive role of various prebiotics as a powerful antioxidant component in aquatic animal diets, the action of prebiotics on fish in terms of antioxidation defense mechanism still seems not clear, and the optimum effective level was not determined [79].

Nonspecific immunity plays a vital role in fish body defense [80]. AKP is an extracellular enzyme involved in the metabolic functions of some diseases, such as growth, cell differentiation, protein synthesis, absorption and transport, and nonspecific immunity [81]. Our results revealed that all five prebiotics could significantly increase the AKP activity of hybrid grouper. The AKP stimulating effect of various prebiotics observed in the present study was consistent with many previous studies, such as Zhang et al. (2021)’s study of FOS in goldfish (*Carassius auratus*), Cao et al. (2019)’s study of GLU in Pengze crucian carp, Zhou et al. (2010)’s study of GLU in red drum (*Sciaenops ocellatus*), and Poolsawat et al. (2021)’s research of XOS in tilapia (*O. niloticus × O. aureus*) [45,46,54,71]. LZM is a critical protective factor in the humoral immune response. LZM can hydrolyze the connection between N-acetyl intracytoic acid and N-acetyl glucosamine, thus destroying the cell wall structure of pathogen microorganisms [82]. Similar to AKP, the LZM activity of hybrid grouper in the present study was also significantly enhanced by all five prebiotics, which is also following many other prebiotic studies. For example, MOS was found to significantly increase the LZM activity of hybrid grouper and European eel (*Anguilla anguilla*) [27,83]; CTS and XOS were reported to increase the LZM activity of Nile tilapia (*Oreochromis niloticus*) significantly [84]. All five prebiotics could induce a significant increase in immune parameters within as short as 4 weeks. The fast immune-regulating effect of prebiotics was also reported in some other prebiotic studies: 0.2% MOS could significantly increase LZM activity in channel catfish (*Ictalurus punctatus*) after 1 week of supplementation [52]; 1% MOS also significantly increased LZM activity and extracellular superoxide anion production in red drum after 4 weeks [85]. Additionally, 0.1% GLU-treated Nile tilapia showed significantly increased LZM activity at the end of the second week and fourth week [44,51]. On the other hand, a variety of prebiotics (e.g., CTS, XOS, and FOS) could induce significant increases of AKP and LZM within as short as 4 weeks, which was encouraging for prebiotic application strategy research since shorter supplementation time would mean less cost of prebiotics and manpower. Another similar “less is more” result was seen in Welker et al. (2012)’s study of Nile tilapia, who found that feeding fish GLU for four weeks and switching to the basal diet caused a significant increase in the respiratory burst of polymorphonuclear lymphocytes compared to fish fed the control diet or the GLU diet continuously [52]. In the future, more short-term (4 weeks or less) experiments should be conducted to investigate whether “less is more” is a common phenomenon among different prebiotics.

Morphological integrity is essential for maintaining the normal functions of the intestine [86]. Intestinal morphology parameters are indicative of healthy gut in fish [82]. Dietary prebiotics can improve the intestinal function of cultured fish, help improve feed utilization, and improve the health status of intestinal mucosal epithelium [25]. In our study, MOS exhibited the best overall intestinal structure-improving effect, significantly enhancing all three intestinal parameters (villus length, villus width, and muscle thickness). XOS and CTS significantly increased two intestinal parameters (villus length and muscle thickness) compared with the control group, which is similar to previous studies in which MOS increased the length of the microchorionic membrane of subadult trout [87], and MOS increased the height of the intestinal villi of redfish [54].

During the aquaculture process, farmed fishes face a variety of stressors. These stressors include deterioration of water quality resulting from heavy rainfall, crowding stress during fish catch/transportation, and so on [5,6,7]. Ammonia is a major environmental stressor for aquatic species [88]. Previous studies have shown that excessive ammonia nitrogen can increase ROS, induce oxidative stress, affect the immune system, and cause organ damage leading to fish death [89]. Furthermore, crowding can cause a stress response, which deteriorates fish health, leading to immunosuppression, increased susceptibility to diseases, and decreased fish survival rate [90,91,92,93,94]. Prebiotics can activate the antioxidant and immune systems of fish, thus significantly improving the survival of aquaculture animals under stressful conditions [88]. This study showed that dietary MOS and XOS could significantly improve the survival of hybrid grouper against ammonia nitrogen stress and crowding stress. Since crowding stress decreased the LZM content of fish serum [92], the fact that dietary MOS and XOS can significantly improve the LZM content of fish serum might partially explain why they can improve hybrid grouper’s resistance to crowding stress. However, FOS, GLU, and CTS also significantly up-regulated the LZM content of fish serum. Why they exhibit non-significant or even adverse effects against stress is unknown. In the present study, the pattern in serum parameter changes induced by anti-stress prebiotics (MOS, XOS) did not distinguish from those induced by non-anti-stress prebiotics (FOS, GLU, CTS). Although XOS and FOS showed the same antioxidant parameter regulatory effects (MDA significantly decreased in our study, CAT significantly increased, GSH-PX and SOD non-significantly increased) and almost the same non-specific immunity regulatory effects (AKP and LZM were significantly increased), their effects on fish were different or even opposite; while XOS significantly improved the survival rate of hybrid grouper after ammonia nitrogen/crowding stress, FOS showed non-significant or even adverse effects against stresses. Though MOS and XOS exhibited similar anti-stress effects, the antioxidant and non-specific immunity parameters they regulated were not the same, indicating that the specific mechanisms of MOS and XOS’s anti-stress effects were probably different. In addition, the doses of MOS and XOS in the present study might also play a role in their anti-stress effects. Like MOS, numerous XOS studies have shown that lower doses of XOS often resulted in better growth/immune enhancing results. For example, 0.1% (not 0.2%, 0.4%, 0.6%) dietary XOS group had significantly lower MDA, higher LZM activity, and higher resistance (survival rate) against *Aeromonas hydrophila* infection [43]. In the future, more comprehensive antioxidant/immune-related indexes and wider ranges of doses should be tested to clarify the mechanism of MOS and XOS’ s protection effect against stresses in hybrid grouper.

Numerous studies have shown that diet prebiotics can effectively improve fish’s resistance to pathogenic bacteria [27,43,45,84,95]. In the present study, hybrid grouper fed with MOS and GLU showed higher survival rates after infection with *V. harvey* than the control group. The result is similar to that of Ren et al. (2020)’s study in which MOS can enhance the resistance of grouper to *V. Harvey*, and Zhao et al. (2011)’s study in which GLU enhances the resistance of sea cucumber to *V. splendidus* [27,95]. In this study, the survival rates of the FOS, CTS, and XOS groups were significantly lower than those of the CON group. This result is inconsistent with several studies: Zhang et al. (2014)’s study showed that dietary FOS promoted disease resistance of ovate pompano against *Vibrio vulnificus* [40]; Fadl et al. (2019)’s study showed that dietary CTS promoted disease resistance of Nile tilapia against *Streptococcus agalactiae* [42]; Zhang et al. (2020)’s study revealed that dietary XOS improved disease resistance of grass carp against *Aeromonas hydrophila*, which may be due to the different pathogenicity of different pathogenic bacteria to fish [43]. The exact reason remains to be determined.

## 5. Conclusions

Different prebiotics have different effects on hybrid grouper. This study indicates that 4 weeks of MOS or XOS supplementation can significantly improve the growth performances, anti-oxidation capacity, non-specific immunity, ammonia nitrogen stress resistance, and crowding stress resistance of juvenile hybrid grouper. Though MOS and XOS exhibited similar anti-stress effects, the antioxidant and non-specific immunity parameters they regulated were not the same, indicating that the specific mechanisms of MOS and XOS’s anti-stress effects were probably different. Furthermore, four weeks of MOS supplementation significantly improved the disease resistance of hybrid grouper against *V. harvey.* In summary, many factors, such as prebiotic structure, dosage, supplementation period, fish species, and age, might contribute to the final results of prebiotic supplementation. Systematic research with scientific designs is of high importance in the future to determine the effect of various prebiotics and their optimum supplementation dosage/period at different life stages of hybrid grouper. In addition, experiments testing the ROS production in the fish blood should be carried out in future, so that the mechanism of prebiotics’ anti-oxidative function could be fully illustrated.

## Figures and Tables

**Figure 1 animals-13-00754-f001:**
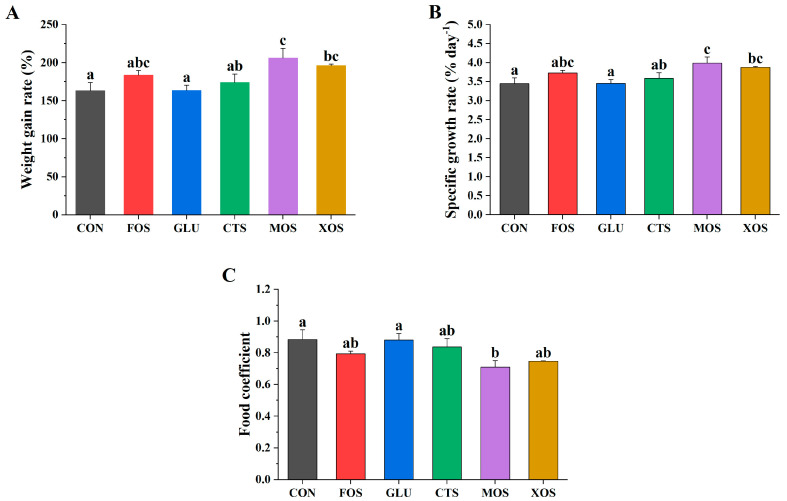
Effects of different prebiotics on growth performance of hybrid grouper. (**A**) WGR, (**B**) SGR, (**C**) FC. Values are expressed as means ± SEM; Mean values in the same row with different superscript letters are significantly different (*p* < 0.05); CON, control; FOS, fructooligosaccharides; GLU, β-glucan; CTS, chitosan; MOS, mannan–oligosaccharides; XOS, xylooligosaccharides; SE, standard error of the mean.

**Figure 2 animals-13-00754-f002:**
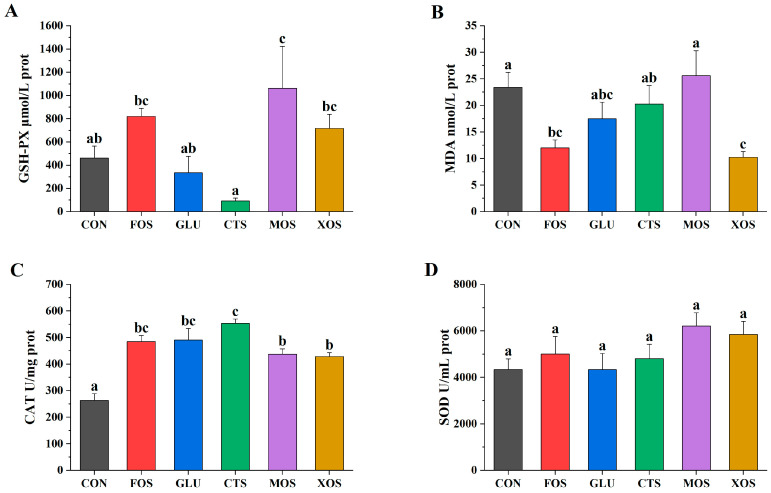
Effects of different prebiotics on antioxidant activities of hybrid grouper. (**A**) GSH-PX, (**B**) MDA, (**C**) CAT, and (**D**) SOD. Values are expressed as means ± SEM; Mean values in the same row with different superscript letters are significantly different (*p* < 0.05); CON, control; FOS, fructooligosaccharides; GLU, β-glucan; CTS, chitosan; MOS, mannan-oligosaccharides; XOS, xylooligosaccharides; SE, standard error of the mean.

**Figure 3 animals-13-00754-f003:**
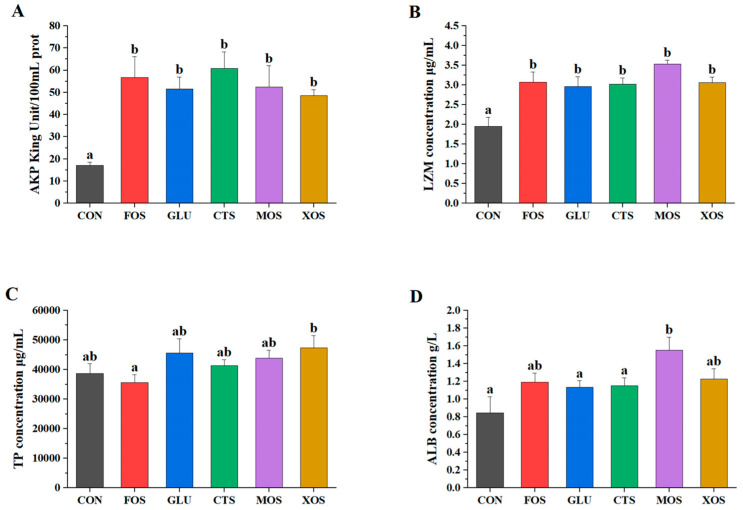
Effects of different prebiotics on non-specific immunity parameters of hybrid grouper. (**A**) AKP, (**B**) LZM, (**C**) TP, and (**D**) ALB. Values are expressed as means ± SEM; Mean values in the same row with different superscript letters are significantly different (*p* < 0.05). CON, control; FOS, fructooligosaccharides; GLU, β-glucan; CTS, chitosan; MOS, mannan–oligosaccharides; XOS, xylooligosaccharides; SE, standard error of the mean.

**Figure 4 animals-13-00754-f004:**
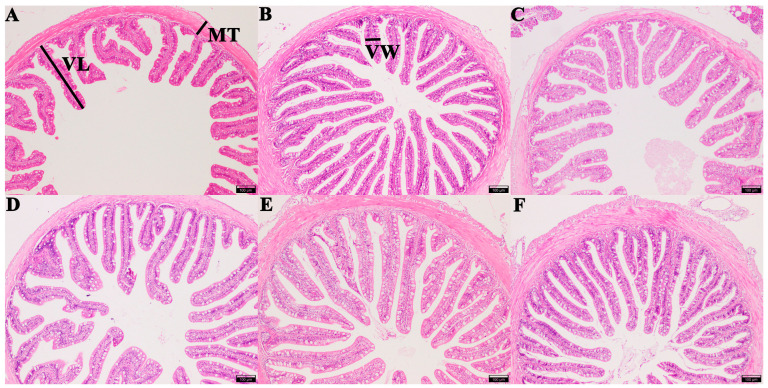
Histological examination images of the intestine in hybrid grouper-fed diets containing different prebiotics (magnification × 100). (**A**) CON; (**B**) FOS; (**C**) GLU; (**D**) CTS; (**E**) MOS; (**F**) XOS. Measured intestinal morphological parameters: villus length (VL), villus width (VW), and muscle thickness (MT). Scale bar: 100 μm. CON, control; FOS, fructooligosaccharides; GLU, β-glucan; CTS, chitosan; MOS, mannan–oligosaccharides; XOS, xylooligosaccharides.

**Figure 5 animals-13-00754-f005:**
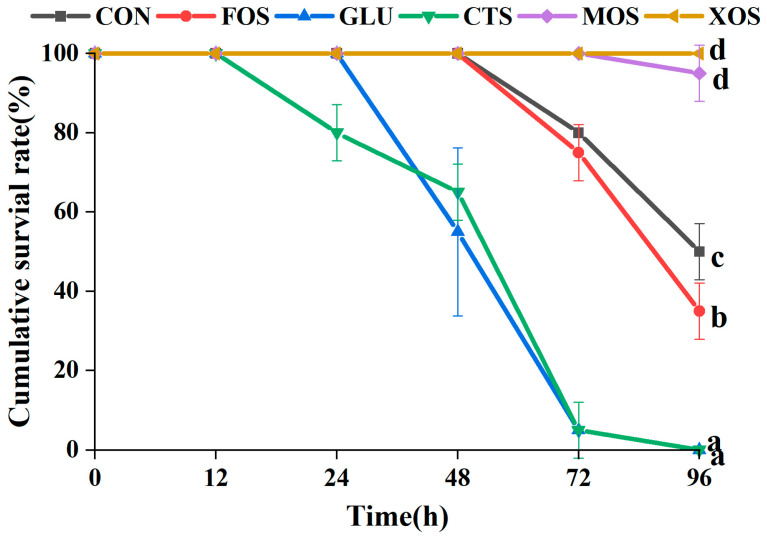
Effects of different prebiotics on the survival rate of hybrid grouper after 96 h ammonia nitrogen stress. Values are expressed as means ± SEM; Mean values at the same time point with different superscript letters are significantly different (*p* < 0.05). CON, control; FOS, fructooligosaccharides; GLU, β-glucan; CTS, chitosan; MOS, mannan–oligosaccharides; XOS, xylooligosaccharides; SE, standard error of the mean.

**Figure 6 animals-13-00754-f006:**
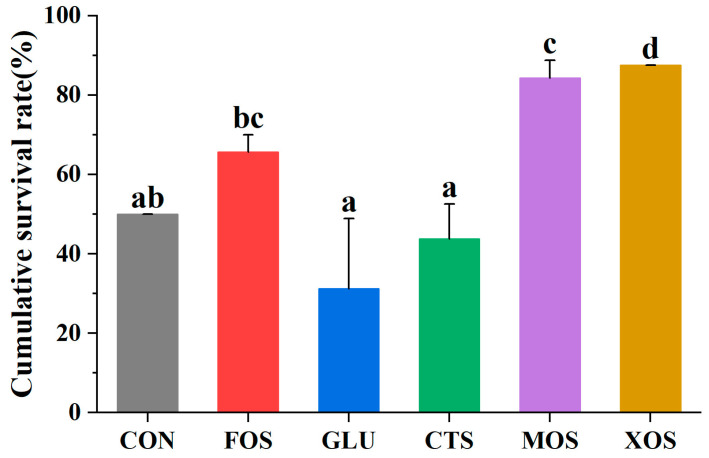
Effects of different prebiotics on the survival rate of hybrid grouper after 24h crowding stress. Values are expressed as means ± SEM; Mean values with different superscript letters are significantly different (*p* < 0.05). CON, control; FOS, fructooligosaccharides; GLU, β-glucan; CTS, chitosan; MOS, mannan-oligosaccharides; XOS, xylooligosaccharides; SE, standard error of the mean.

**Figure 7 animals-13-00754-f007:**
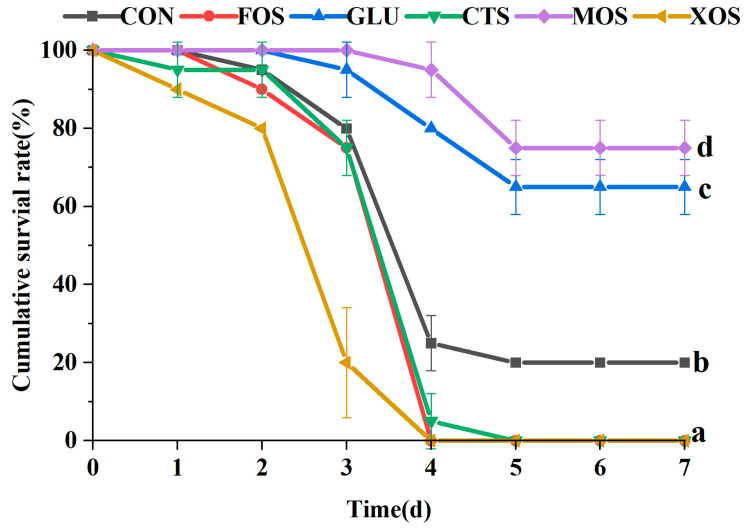
Effects of different prebiotics on the cumulative survival rate of hybrid grouper after challenging with *V. harvey*. Values are expressed as means ± SEM; Values at the same time point with different superscript letters are significantly different (*p* < 0.05). CON, control; FOS, fructooligosaccharides; GLU, β-glucan; CTS, chitosan; MOS, mannan–oligosaccharides; XOS, xylooligosaccharides; SE, standard error of the mean.

**Table 1 animals-13-00754-t001:** Ingredients of the experimental diets.

	Experimental Diets					
Ingredients (g/kg Diet)	CON	CTS	GLU	FOS	MOS	XOS
Common ingredients ^a^	1000	995	999	998	998	999.5
Chitosan ^b^	0	5	0	0	0	0
β-glucan ^b^	0	0	1	0	0	0
Fructooligosaccharides ^b^	0	0	0	2	0	0
Mannooligosaccharides ^b^	0	0	0	0	2	0
Xylooligosaccharides ^b^	0	0	0	0	0	0.5

^a^ Common ingredients (g/kg diet): Grouper feed (Guangdong Hengxing Aquatic Products Co., Ltd., Zhanjiang, China). ^b^ Purchased from Shandong Shengyuan Biotechnology Co., Ltd., Qingdao, China.

**Table 2 animals-13-00754-t002:** Intestinal morphology of hybrid grouper fed with different prebiotics.

Group	Villi Length (μm)	Villi Width (μm)	Muscle Thickness (μm)
CON	285.37 ± 14.69 ^a^	42.57 ± 4.25 ^a^	42.87 ± 6.06 ^a^
CTS	403.57 ± 35.03 ^b^	45.4 ± 4.61 ^a^	66.8 ± 7.05 ^bc^
GLU	259.01 ± 15.82 ^a^	50.03 ± 0.78 ^ab^	74.87 ± 4.87 ^cd^
FOS	425.23 ± 2.59 ^b^	37.47 ± 0.77 ^a^	54.03 ± 2.49 ^ab^
MOS	421.4 ± 21.89 ^b^	64.1 ± 9.91 ^b^	89 ± 5.23 ^d^
XOS	414.83 ± 12.35 ^b^	38.83 ± 1.28 ^a^	119.03 ± 8.22 ^e^

Values are expressed as means ± SEM; Mean values in the same column with different superscript letters are significantly different (*p* < 0.05); CON, control; FOS, fructooligosaccharides; GLU, β-glucan; CTS, chitosan; MOS, mannan-oligosaccharides; XOS, xylooligosaccharides; SE, standard error of the mean.

## Data Availability

The data that support the findings of this study are available from the authors upon reasonable request.

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
