# Peer review of "Effects of Five Prebiotics on Growth, Antioxidant Capacity, Non-Specific Immunity, Stress Resistance, and Disease Resistance of Juvenile Hybrid Grouper (Epinephelus fuscoguttatus ♀ × Epinephelus lanceolatus ♂)"

_animals, 2023, doi:10.3390/ani13040754_

Round 1
Reviewer 1 Report
Dear editor,
Firstly, I would like to thank the opportunity to review the work entitled “Effects of Five Prebiotics on Growth, Antioxidant Capacity, Non-specific Imnity, Stress Resistance, and Disease Resistance of Juvenile Hybrid Grouper (Epinephelus fuscoguttatusmâ™€× Epinephelus lanceolatus♂)” by Zhu and colleagues.
Secondly, I would like to show my overall enthusiasm for this work. I recommend it for publication after some improvements are performed. You will find below my comments to the authors.
Dear authors,
Please should note that prebiotics and other means of supplementation, should be used carefully during specific periods of fish life. If they are used in the medium/long term, implications on energetic balances and permanent alterations in other biological processes may appear. The prebiotic systematic use will not favour the fish growth and therefore the economic profitability of the fish farming businesses. They may also increase unnecessarily the production costs. From this, I suggest adding a clear sentence on the limitations of prebiotics in the introduction section.
Specific examples of stressful situations where the use of prebiotics is useful can be provided (prior to the transport of fish; prior to treatments; etc.) from the bibliography. Indeed, the authors have made effort to show results in different domains that may not seem fully explored. For example, MOS seemed great for protecting grouper against some bacteria; while XOS is better for protecting “oxidative stressful” situations.
It is strange for me to understand how sustainability and improvement of health in long term can result from the inclusion of prebiotics in feeds. Please be aware that water quality, disease prevention and else, should always be provided, and producers should not rely on prebiotics to solve “deep (infra-)structural” problems! I would suggest to re-write some sections related to this view of prebiotics as solutions for severe problems (ex.: Lines 51/53/55; 411).
Lines 94 and 101 add the reasons why such doses were used. Where based on optimized results from previous works? Same species? Were wide ranges tested for all pre-biotics before?
Line 109 Please provide information regarding water recirculation/renewal conditions, water quality control and water treatments if any.
Figure 6 Please confirm error bars in control and XOS treatments.
Line 291 please simplify or divide the sentence, there are three words “different”.
Line 294 Please re-order the sentence, or eventually delete the example of tiger prawn since it is "biologically" more distant from grouper than turbot
Line 308 Is pre-biotic supplementation age-related? Or can it be related to fish age/stage/weight? Please add further discussion on this issue since this is one important conclusion to be further studied.
Line 318 Please break the paragraph here, starting a new one in “Dosage…”
Line 322 “Many factors… “ this sentence here has great importance, so it should be emphasized in the introduction (and conclusions)
Line 435 Further discussion should be added regarding the MOS and XOS doses tested. Would lower testing doses help the results of MOS?
Line 437 This is also an important conclusion to be added to the conclusions section (and abstract if possible)
Line 461 I am not sure that I understand the meaning of this sentence, please rephrase
Author Response
Point 1: Please should note that prebiotics and other means of supplementation, should be used carefully during specific periods of fish life. If they are used in the medium/long term, implications on energetic balances and permanent alterations in other biological processes may appear. The prebiotic systematic use will not favour the fish growth and therefore the economic profitability of the fish farming businesses. They may also increase unnecessarily the production costs. From this, I suggest adding a clear sentence on the limitations of prebiotics in the introduction section.
Response 1: We have added sentences on the limitations of prebiotics in the introduction section. It was located in the second paragraph of Introduction section. (Line 91-94 in revised manuscript)
Point 2: Specific examples of stressful situations where the use of prebiotics is useful can be provided (prior to the transport of fish; prior to treatments; etc.) from the bibliography. Indeed, the authors have made effort to show results in different domains that may not seem fully explored. For example, MOS seemed great for protecting grouper against some bacteria; while XOS is better for protecting “oxidative stressful” situations.
Response 2: We gave specific examples of stressful situations where prebiotics would be useful. (Line 59-62,97-98 in our revised manuscript)
Point 3: It is strange for me to understand how sustainability and improvement of health in long term can result from the inclusion of prebiotics in feeds. Please be aware that water quality, disease prevention and else, should always be provided, and producers should not rely on prebiotics to solve “deep (infra-)structural” problems! I would suggest to re-write some sections related to this view of prebiotics as solutions for severe problems (ex.: Lines 51/53/55; 411).
Response 3: We agree with our reviewer that inclusion of prebiotics in feeds alone cannot solve the complex problems in aquaculture. We have rewritten the relative sentences. (Line 59-65, 449-451 in our revised manuscript)
Point 4: Lines 94 and 101 add the reasons why such doses were used. Where based on optimized results from previous works? Same species? Were wide ranges tested for all pre-biotics before?
Response 4: Thanks for our reviewer’s suggestion. We have added the reasons.(Line 123-125 in our revised manuscript)
Point 5: Line 109 Please provide information regarding water recirculation/renewal conditions, water quality control and water treatments if any.
Response 5: Following our reviewer’s suggestion, we have added the relative information (Line 144-148 in our revised manuscript)
Point 6: Figure 6 Please confirm error bars in control and XOS treatments.
Response 6: We went back and checked our raw data, and found that the survival rates of the parallel experimental groups in CON and XOS treatments were indeed the same. Therefore, the values of the error bars would be zero for CON and XOS group. We re-drew Figure 6, so that the error bars (values = 0) of CON and XOS groups were more evident. (Line 316 in our revised manuscript)
Point 7: Line 291 please simplify or divide the sentence, there are three words “different”."
Response 7: We deleted these sentences following another reviewer’s suggestion. (Line 335 in our revised manuscript)
Point 8: Line 294 Please re-order the sentence, or eventually delete the example of tiger prawn since it is "biologically" more distant from grouper than turbot
Response 8: We replaced the tiger prawn example with a Japanese eel example, so that all the examples were about fish. (Line 336-337 in our revised manuscript)
Point 9: Line 308 Is pre-biotic supplementation age-related? Or can it be related to fish age/stage/weight? Please add further discussion on this issue since this is one important conclusion to be further studied.
Response 9: We added relative discussion here. (Line 360-364 in our revised manuscript)
Point 10: Line 318 Please break the paragraph here, starting a new one in “Dosage…”
Response 10: We broke the paragraph and started a new paragraph in “dosage...”. (Line 356 in our revised manuscript)
Point 11: Line 322 “Many factors… “ this sentence here has great importance, so it should be emphasized in the introduction (and conclusions)
Response 11: Following our reviewer’s suggestion, we emphasized this sentence both in the introduction and conclusions. (Line 89-91, 509-511 in our revised manuscript)
Point 12: Line 435 Further discussion should be added regarding the MOS and XOS doses tested. Would lower testing doses help the results of MOS?
Response 12: We added further discussion regarding the MOS and XOS doses. (Line 477-482 in our revised manuscript)
Point 13: Line 437 This is also an important conclusion to be added to the conclusions section (and abstract if possible)
Response 13: We have added the sentences “Though MOS and XOS exhibited similar anti-stress effects... MOS and XOS’s anti-stress effects were probably different.” to both the conclusions and abstract. (Line 43-45, 504-507 in our revised manuscript)
Point 14: Line 461 I am not sure that I understand the meaning of this sentence, please rephrase
Response 14: It was an editing mistake. The sentence should be deleted. We have deleted the sentence.
Reviewer 2 Report
The authors should include the table of the experimental diets. After that it may be more easy to give a general comment of the experimental plan.
Authors should describe better the diets with their ingredients. In the conclusion, they should underline the need of further experiments to test the ROS production in the fish blood.
Author Response
Point 1: The authors should include the table of the experimental diets. After that it may be more easy to give a general comment of the experimental plan.
Response 1: Following our reviewer’s suggestion, we included the following table to better explain our experimental plan. (Line 133 in our revised manuscript)
Table 1. Ingredients of the experimental diets.
|
|
Experimental diets |
|||||
|
Ingredients (g/kg diet) |
CON |
CTS |
GLU |
FOS |
MOS |
XOS |
|
Common ingredientsa |
1000 |
995 |
999 |
998 |
998 |
999.5 |
|
Chitosanb |
0 |
5 |
0 |
0 |
0 |
0 |
|
β-glucanb |
0 |
0 |
1 |
0 |
0 |
0 |
|
Fructooligosaccharidesb |
0 |
0 |
0 |
2 |
0 |
0 |
|
Mannooligosaccharidesb |
0 |
0 |
0 |
0 |
2 |
0 |
|
Xylooligosaccharidesb |
0 |
0 |
0 |
0 |
0 |
0.5 |
a Common ingredients (g/kg diet): Grouper feed (Guangdong Hengxing Aquatic Products Co., Ltd., China).
b Purchased from Shandong Shengyuan Biotechnology Co., Ltd., China.
Point 2: Authors should describe better the diets with their ingredients. In the conclusion, they should underline the need of further experiments to test the ROS production in the fish blood.
Response 2: We added a table listing the ingredients of different diets. And we added the following sentences in Conclusion to underline the need of further experiments to test the ROS production in the fish blood.
“In addition, experiments testing the ROS production in the fish blood should be carried out in future, so that the mechanism of prebiotics’s anti-oxidative function could be fully illustrated.” (Line 133-136, 513-515 in our revised manuscript)
Reviewer 3 Report
MS: Effects of five prebiotics on growth, antioxidant capacity, non-specific immunity, stress resistance, and disease resistance of juvenile hybrid grouper (Epinephelus fuscoguttatusmâ™€× Epinephelus lanceolatus♂)
Manuscript ID: animals-2190902
Summary:
The authors did an interesting work about the impact of several prebiotics on antioxidant and non-specific immune parameters, and on midgut morphology and stress and disease resistance, in Juvenile Hybrid Grouper. The authors have found clear evidence that two prebiotics, MOS and XOS, had the most beneficial results, by increasing the antioxidant activities and the stress and infection by Vibrio Harvey resistance. I believe this preliminary results might contribute to overcome the problems associated with the aquaculture of Hybrid Grouper. Nevertheless, the authors should demonstrate how this study contributes to the field and stands out from the other studies.
General concept comments:
The manuscript is well written, well structured and very clear about the methods and the results obtained. However, two major issues are of concern and should be improved, the introduction and the discussion.
Abstract: The results obtained for histomorphometric analyses of the midgut should be included.
Introduction: The introduction is well structured and provides the necessary background to understand the problematic of the study. The authors have stated that little information about the effects of prebiotics on Hybrid Grouper is available (L83) but previously (L61-78) the authors provided a characterization of the general effects of the prebiotics tested in this study on other fish, and also in hybrid grouper. There is in fact some information, including the papers cited by the authors (one of the same group) and other papers not cited (and were published recently). It is necessary to clarify in the introduction how this study is different from the others and what is the contribution to the field. A simple sentence would be enough with the necessary improvement of these two last paragraphs of the introduction.
Discussion: The discussion can be considerably shortened as there are sentences that does not add anything to the comprehension of the results or are difficult to understand the purpose. I suggest that the authors open up the discussion with the main finding of this study. What was the prebiotic with most change in the immune related parameters and disease/stress resistance? The discussion can be developed from this point. Then the authors could provide a discussion about the results obtained for MDA and the antioxidant enzymes. Why all groups had increased catalase, but MDA was only reduced in FOS and XOS? Regarding the intestine morphology, why do the authors consider that the increase in the midgut morphological parameters is an improvement? Overall, an increase in villi heigh and width is considered a sign of intestinal inflammation. This part of the discussion needs a bit more consideration.
Figures: Figures 1 to 4 are out of focus. Specifically, figure 4 has poor quality images. Also, I cannot see the morphological differences detected by histomorphometric analyses. There is almost no difference in villi width between A and E. Authors should provide focused images and it would be easier to have the same villi orientation in all the images.
Specific comments:
L23: Six groups, not seven.
L29: Was significantly enhanced in the MOS group compared to what group?
L77-88: This should be rewritten based on the comments provided previously.
L136: For antioxidant activities, and non-specific immune parameters as well, information about the absorbance reading equipment and if the reactions were performed in 96 wells should be included.
L153: I don’t understand the reference to 45. Also, it is fundamental to include more information about how and how many measurements were performed for each histological parameter.
L208 (Figure 1C): CON and GLU should be a, and MOS should be b.
L219: I suggest to remove the sentence. Just state that they are not significantly different as this sentence adds confusion.
L221 (Figure 2B): Reconsider the c and a letters. It is strange to see c in the control when it should be a (the first letter of the alphabet).
L230-232: Only MOS is significantly different from CON.
L253: It would be easier to follow if authors provide the name of the group diets, not the prebiotic, following the same rationale as the other captions.
L273 (Figure 6): Please confirm if XOS is significantly different form MOS.
L291-293: I tis difficult to understand what authors are trying to say with this sentence. I suggest to remove it as it does not add anything meaningful to the discussion.
L299-322: I think this part of the discussion is too long and confusing.
L336: What does it mean "certain levels"?
L338-340: References?
L388-389: I don’t understand this sentence.
L408-410: I suggest to remove from the discussion what is not significant and thus not add anything meaningful to the understanding of the results.
L421-422: Out of place.
L426-429: I don’t understand this part.
L430-431: Up our down-regulated should be replaced by increased or decreased activities.
L461-463: The conclusion should be focused on your own results.
L463-465: Difficult to understand the meaning of this sentence.
Author Response
Point 1: Abstract: The results obtained for histomorphometric analyses of the midgut should be included.
Response 1: We added the histomorphometric analyses results as our reviewer suggested. The sentences are as follows: “Histological examination of the intestine revealed that muscle thickness was significantly increased in all prebiotic supplemented groups compared to the CON group (P < 0.05). Villi length, villi width, muscle thickness all increased significantly in MOS group (P < 0.05).”(Line 38-41 in our revised manuscript)
Point 2: Introduction: The introduction is well structured and provides the necessary background to understand the problematic of the study. The authors have stated that little information about the effects of prebiotics on Hybrid Grouper is available (L83) but previously (L61-78) the authors provided a characterization of the general effects of the prebiotics tested in this study on other fish, and also in hybrid grouper. There is in fact some information, including the papers cited by the authors (one of the same group) and other papers not cited (and were published recently). It is necessary to clarify in the introduction how this study is different from the others and what is the contribution to the field. A simple sentence would be enough with the necessary improvement of these two last paragraphs of the introduction.
Response 2: Following our reviewer’s suggestion, we added sentences regarding “how this study is different from the others and what is the contribution to the field” in the last two paragraphs of introduction section.
The added sentences are: “Therefore, researches regarding comparison of various prebiotic’s specific beneficial effects are necessary to identify the optimal prebiotic...” and “Studies regarding prebiotics’ effects on this species were few. Besides two MOS studies in hybrid grouper (both from our lab), there is very limited information concerning how other prebiotics might affect this fish species.” (Line 95-98, 101-104 in our revised manuscript)
Point 3: Discussion: The discussion can be considerably shortened as there are sentences that does not add anything to the comprehension of the results or are difficult to understand the purpose. I suggest that the authors open up the discussion with the main finding of this study. What was the prebiotic with most change in the immune related parameters and disease/stress resistance? The discussion can be developed from this point. Then the authors could provide a discussion about the results obtained for MDA and the antioxidant enzymes. Why all groups had increased catalase, but MDA was only reduced in FOS and XOS? Regarding the intestine morphology, why do the authors consider that the increase in the midgut morphological parameters is an improvement? Overall, an increase in villi heigh and width is considered a sign of intestinal inflammation. This part of the discussion needs a bit more consideration.
Response 3: Following our reviewer’s suggestion, we re-wrote many sentences in the discussion section.(Line 378-385 in our revised manuscript)
As for the discussion of serum parameters, we did try to combine the antioxidant parameters with non-specific immune parameters, and tried to correlate the changes of these parameters to the stress/disease resistance effects of prebiotics. But the only correlation we could find was: MOS/XOS exhibited the most changes in non-specific immune parameters (and in serum parameters taken together), and MOS/XOS were the only prebiotics with significant ammonia/crowding stress resistance effects. But afterwards, it’s really hard to explain why disease resistance effects were seen in only MOS and GLU (not XOS, not FOS). After all, XOS showed the largest changes in both non-specific immune (AKP↑, LZM↑, TP↑) and antioxidant parameters (CAT↑, MDA↓). Why XOS decrease the disease resistance of fish? FOS is even more confusing. FOS showed the second most changes in serum parameters (AKP↑, LZM↑, CAT↑, MDA↓), why it had no effect against stress and had adverse effect against disease?
For above reasons, we maintained the original discussion on serum parameters. But we did tried to condense the sentences in this part of discussion. We also added the discussion concerning the relationship between CAT and MDA. The following are the added sentences:
“In the present study, all prebiotic-supplemented groups had increased CAT activities, but MDA was only reduced in FOS and XOS groups. The result indicated that CAT was not always negatively correlated with MDA. This is consistent with many other studies. [63,64,65,66]. The explanation is rather obvious: After all, CAT is just one of numerous antioxidant enzymes that could prevent the production of MDA. Besides antioxidant enzymes, a large number of other antioxidants, such as non-enzymatic molecules (e.g. vitamin C, E), certain prebiotics/probiotics, could also affect the final MDA level of fish[67,68,69].”(Line 378-385 in our revised manuscript)
As for the relationship between villi height/width and intestinal inflammation, scientists don’t have an agreement right now. We searched available literature (aquatic animals, mammal/human), and noticed considerable disagreements concerning interpretation of intestinal morphological changes. Of the five prebiotics, MOS exhibited the best overall growth promoting, anti-stress and anti-disease effects. The morphological parameters in MOS group also exhibited the most changes (villus length↑, villus width↑, muscle thickness↑). If these changes all imply intestinal inflammation (which is often the result of infection/intestinal damage), how to explain the beneficial effects of MOS? For the above reasons, we didn’t change the discussion concerning the morphological parameter changes.
Table 1. Summary of different opinions and related references.
|
Opinion |
References |
|
Opinion 1: Villus length/width, muscle thickness increases imply intestinal inflammation |
Dongmei Zhang, Liulan Zhao, Qishuang He, Ahmed Abdi Adam, Kuo He, Lisen Li, Xin Zhang, Jie Luo, Wei Luo, Zhiqiong Li, Song Yang, Qiao Liu, Intermittent hypoxia exposure alleviates 2,4,6-trinitrobenzene sulfonic acid-induced enteritis by enhancing the intestinal barrier and inhibiting endoplasmic reticulum stress in juvenile largemouth bass, Aquaculture,Volume 563, Part 1,2023,738951, https://doi.org/10.1016/j.aquaculture.2022.738951. |
|
Shengming Sun, Yanli Su, Han Yu, Xianping Ge, Chengfeng Zhang,Starvation affects the intestinal microbiota structure and the expression of inflammatory-related genes of the juvenile blunt snout bream, Megalobrama amblycephala,Aquaculture,Volume 517,2020,734764, https://doi.org/10.1016/j.aquaculture.2019.734764. |
|
|
Pereira JNB, Murata GM, Sato FT, Marosti AR, Carvalho CRO, Curi R. Small intestine remodeling in male Goto-Kakizaki rats. Physiol Rep. 2021 Feb;9(3):e14755. doi: 10.14814/phy2.14755. |
|
|
Opinion 2: Villus length/width, muscle thickness increases imply improvement of intestinal health |
Roberta Imperatore, Graziella Orso, Serena Facchiano, Pierpaolo Scarano, Seyed Hossein Hoseinifar, Ghasem Ashouri, Carmine Guarino, Marina Paolucci, Anti-inflammatory and immunostimulant effect of different timing-related administration of dietary polyphenols on intestinal inflammation in zebrafish, Danio rerio, Aquaculture,Volume 563, Part 1,2023,738878, https://doi.org/10.1016/j.aquaculture.2022.738878. |
|
Muzi Zhang, Haibo Jiang, Shidong Wang, Ge Shi, Ming Li, Effects of dietary cellulose supplementation on the intestinal health and ammonia tolerance in juvenile yellow catfish Pelteobagrus fulvidraco,Aquac. Rep.Volume 28,2023,101429, https://doi.org/10.1016/j.aqrep.2022.101429. |
|
|
Yajun Hu, Junzhi Zhang, Junjing Xue, Wuying Chu, Yi Hu, Effects of dietary soy isoflavone and soy saponin on growth performance, intestinal structure, intestinal immunity and gut microbiota community on rice field eel (Monopterus albus), Aquaculture,Volume 537,2021,736506, https://doi.org/10.1016/j.aquaculture.2021.736506. |
|
|
Park, M.; Park, E.-J.; Kim, S.-H.; Lee, H.-J. Lactobacillus plantarum ATG-K2 and ATG-K6 Ameliorates High-Fat with High-Fructose Induced Intestinal Inflflammation. Int. J. Mol.Sci. 2021, 22, 4444. https://doi.org/10.3390/ijms22094444 |
|
|
Liu, Y., Choe, J., Kim, S. et al. Dietary spray-dried plasma improves intestinal morphology of mated female mice under stress condition. J Anim Sci Technol 60, 10 (2018). https://doi.org/10.1186/s40781-018-0169-5 |
|
|
Zhou, Q.; Buentello, J. A.; Gatlin, D. M. Effects of dietary prebiotics on growth performance, immune response and intestinal morphology of red drum (Sciaenops ocellatus). Aquaculture 2010, 309, (1), 253-257. |
|
|
Hahor, W.; Thongprajukaew, K.; Suanyuk, N. Effects of dietary supplementation of oligosaccharides on growth performance, gut health and immune response of hybrid catfish (Pangasianodon gigas × Pangasianodon hypophthalmus). Aquaculture 2019, 507, 97-107. |
|
|
Opinion 3: Both Opinion 1&2 might be right. Conflicting results exist. |
Peuhkuri K, Vapaatalo H, Korpela R. Even low-grade inflamma tion impacts on small intestinal function. World J Gastroenterol 2010; 16(9): 1057-1062 DOI: http://dx.doi.org/10.3748/wjg.v16.i9.1057
|
Point 4: Figures: Figures 1 to 4 are out of focus. Specifically, figure 4 has poor quality images. Also, I cannot see the morphological differences detected by histomorphometric analyses. There is almost no difference in villi width between A and E. Authors should provide focused images and it would be easier to have the same villi orientation in all the images.
Response 4: Thanks for our reviewer’s comments. We changed the images following our reviewer’s suggestion. Hopefully now they can meet the requirement (Line 250,263,293 in our revised manuscript).
Point 5: L23: Six groups, not seven.
Response 5: Thank you. We corrected the mistake (Line 24 in our revised manuscript).
Point 6: L29: Was significantly enhanced in the MOS group compared to what group?
Response 6: Compared to the CON group. The sentence has been revised to “MOS and XOS significantly improved the growth of hybrid grouper compared to the CON group (P < 0.05).” (Line 29 in our revised manuscript)
Point 7: L77-88: This should be rewritten based on the comments provided previously.
Response 7: It has been rewritten following our reviewer’s suggestion. Please refer to Response 2. (Line 95-98, 101-104 in our revised manuscript)
Point 8: L136: For antioxidant activities, and non-specific immune parameters as well, information about the absorbance reading equipment and if the reactions were performed in 96 wells should be included.
Response 8: Following our reviewer’s suggestion, we added the following information “All reactions were performed in 96 wells, and the absorbance was read by a microplate reader (Thermo Scientific™ 5580, Shanghai, China)” (Line 179-180,190-191 in our revised manuscript) .
Point 9: L153: I don’t understand the reference to 45. Also, it is fundamental to include more information about how and how many measurements were performed for each histological parameter.
Response 9: The reference to 45 was unnecessary. We deleted “[45]”. For calculating the mean villus height and thickness for each fish at least twenty measurements were done. For the measurements, 20 images per individual (4 quadrants/section × 5 sections/individual) was captured and measured. Each intestinal histological parameter was calculated based on 20 measurements per individual (Line 198-201 in our revised manuscript) .
Point 10: L208 (Figure 1C): CON and GLU should be a, and MOS should be b.
Response 10: We agree with our reviewer. We have revised the figure accordingly (Line 250 in our revised manuscript).
Point 11: L219: I suggest to remove the sentence. Just state that they are not significantly different as this sentence adds confusion.
Response 11: We agree with our reviewer. That sentence has been shortened as “The SOD activities of all prebiotic groups were not significantly different from that of the CON group (P > 0.05)”.(Line 261-262 in our revised manuscript)
Point 12: L221 (Figure 2B): Reconsider the c and a letters. It is strange to see c in the control when it should be a (the first letter of the alphabet).
Response 12: Thanks for our reviewer’s comments. We have changed the letters, and used “a” for control group (Line 263 in our revised manuscript).
Point 13: L230-232: Only MOS is significantly different from CON.
Response 13: We have revised the sentence following our reviewer’s suggestion (Line 274 in our revised manuscript).
Point 14: L253: It would be easier to follow if authors provide the name of the group diets, not the prebiotic, following the same rationale as the other captions.
Response 14: We agree with our reviewer. We have made the corresponding revision. Prebiotic abbreviations were used instead of full names (Line 297-298 in our revised manuscript).
Point 15: L273 (Figure 6): Please confirm if XOS is significantly different form MOS.
Response 15: Yes, XOS is significantly different from MOS. The error bars of XOS and CON columns were not clearly shown before. We redrew the figure, and made them more evident (Line 316 in our revised manuscript).
Point 16: L291-293: I tis difficult to understand what authors are trying to say with this sentence. I suggest to remove it as it does not add anything meaningful to the discussion.
Response 16: Following our reviewer’s suggestion, the sentences were removed (Line 335 in our revised manuscript).
Point 17: L299-322: I think this part of the discussion is too long and confusing.
Response 17: We rewrote the paragraph, trying to be shorter and clearer (Line 341-359 in our revised manuscript).
Point 18: L336: What does it mean "certain levels"?
Response 18: “Certain levels” were changed to “various levels” (Line 386 in our revised manuscript).
Point 19: L338-340: References?
Response 19: We added the corresponding references (Line 389-391 in our revised manuscript).
Point 20: L388-389: I don’t understand this sentence.
Response 20: The sentence was deleted in revised manuscript. We were trying to say: A number of prebiotic studies reached conflicting results regarding the effects of prebiotic on non-specific immunity. But now we realized this sentence is not related to this paragraph, so we deleted it (Line 429 in our revised manuscript).
Point 21: L408-410: I suggest to remove from the discussion what is not significant and thus not add anything meaningful to the understanding of the results.
Response 21: We agree with our reviewer. Non-significant results don't need to be discussed. We have deleted the sentence (Line 448 in our revised manuscript).
Point 22: L421-422: Out of place.
Response 22: We re-wrote the sentence, so that it fit in the context (Line 461 in our revised manuscript).
Point 23: L426-429: I don’t understand this part.
Response 23: We re-wrote this part. “The pattern in serum parameter changes induced by anti-stress prebiotics (MOS & XOS) didn’t distinguish from those induced by non-anti-stress prebiotics (FOS, GLU, CTS)” (Line 466-468 in our revised manuscript).
Point 24: L430-431: Up our down-regulated should be replaced by increased or decreased activities.
Response 24: We changed “up-regulated” to “increased”, and “down-regulated” to “decreased” (Line 469-472 in our revised manuscript).
Point 25: L461-463: The conclusion should be focused on your own results.
Response 25: Those sentences were deleted (Line 509 in our revised manuscript).
Point 26: L463-465: Difficult to understand the meaning of this sentence.
Response 26: The sentence was replaced by “Systematic research with scientific designs is of high importance in future to determine the effect of various prebiotics and their optimum supplementation dosage/period at different life stages of hybrid grouper.” (Line 511-513 in our revised manuscript)